# Cognitive–Affective Risk Factors of Female Intimate Partner Violence Victimization: The Role of Early Maladaptive Schemas and Strategic Emotional Intelligence

**DOI:** 10.3390/brainsci13071118

**Published:** 2023-07-24

**Authors:** Klaudia Sójta, Aleksandra Margulska, Wioletta Jóźwiak-Majchrzak, Anna Grażka, Karolina Grzelczak, Dominik Strzelecki

**Affiliations:** 1Department of Affective and Psychotic Disorders, Medical University of Lodz, Czechoslowacka Street 8/10, 92-216 Lodz, Poland; klaudia.krakus@stud.umed.lodz.pl (K.S.); anna.bartosiewicz@stud.umed.lodz.pl (A.G.); 2Department of Adolescent Psychiatry, Medical University of Lodz, Czechoslowacka Street 8/10, 92-216 Lodz, Poland; aleksandra.margulska@umed.lodz.pl; 3Department of Applied Sociology and Social Work, University of Lodz, Rewolucji 1905 41/43, 90-214 Lodz, Poland; wioletta.jozwiak.majchrzak@edu.uni.lodz.pl; 4Medical University of Lodz, 92-216 Lodz, Poland; karolinamariagrzelczak@gmail.com

**Keywords:** intimate partner violence, early maladaptive schemas, strategic emotional intelligence, risk factors, victimization

## Abstract

(1) Background: Intimate partner violence (IPV) is a pervasive and destructive phenomenon. There is a need for an integrated and comprehensive approach to IPV in order to align prevention, support and treatment. Still little is known about the cognitive and affective markers of IPV that are modifiable. Such knowledge, therefore, can support the effectiveness of prevention and intervention programs. In this study, we put forward a hypothesis that, after accounting for the influence of sociodemographic variables, the domains of early maladaptive schemas (EMS) and strategic emotional intelligence would provide additional information for predicting female IPV victimization. (2) Methods: 48 female survivors of IPV and 48 age-matched women with no prior experience of IPV completed the Young Schema Questionnaire-Short Form 3 (YSQ-SF3) and The Emotional Understanding Test (TRE). (3) Results: The domains of disconnection and rejection and impaired limits were significant predictors of IPV victimization, but the results did not support the predictive value for impaired autonomy, other-directedness and strategic emotional intelligence. (4) Conclusions: Our findings add to the emerging evidence of a link between disconnection and rejection domain and IPV victimization. As a consequence, maladaptive beliefs that interpersonal relationships are unstable and insecure and expose to the risk of humiliation and harm, and that basic emotional needs cannot be satisfied in close relationships, are associated with a higher risk of intimate partner violence. In this context, schema therapy appears to be a promising support for IPV victims.

## 1. Introduction

Violence against women is a pervasive and destructive phenomenon that is increasingly recognized as a serious public health concern around the world. The Istanbul Convention defines “violence against women” as “a violation of human rights and a form of discrimination against women and shall mean all acts of gender-based violence that result in, or are likely to result in, physical, sexual, psychological or economic harm or suffering to women, including threats of such acts, coercion or arbitrary deprivation of liberty, whether occurring in public or in private life” [1]. The pervasiveness of violence against women is a critical factor in perpetuating gender inequality and is considered a gross violation of fundamental rights with respect to dignity and equality. Prevalence studies conducted by the World Health Organization in 2013 [2] and EU Agency for Fundamental Rights (FRA) in 2014 reveal alarming rates of violence perpetrated against women. In particular, it has been estimated that approximately 30% of the female population have been victims of physical and/or sexual violence at some point since the age of 15. According to an EU-wide survey, an estimated 13 million women in the European Union have experienced physical violence in the 12 months preceding the interviews. In addition, an estimated 3.7 million women in the EU have experienced sexual violence over the same period [3]. Intimate partner violence (IPV) has been found to be associated with a broad spectrum of both short-term and long-term consequences for physical and mental health. Physical health issues included but were not limited to an increased risk of injury [4,5], chronic pain [6,7], headaches [8,9], diabetes [10,11] and higher rates of sexually transmitted infections [12,13,14]. A recent meta-analysis showed an increased likelihood of all considered IPV mental health outcomes, including depression (OR = 2.04–3.14), post-traumatic stress disorder (PTSD) (OR = 2.15–3.14), and suicidality (OR = 2.17–5.52) [15]. The most serious manifestation of IPV is intimate partner homicide, with women disproportionately affected [16]. While the severe harm inflicted upon women who experience violence is evident, the effects of victimization go far beyond the individual level and have repercussions for families and society [17]. Despite international efforts to reduce gender-based victimization, starting with the United Nations General Assembly’s Declaration on the Elimination of Violence Against Women [18], recent reports reveal an alarming increase in IPV since the onset of the COVID-19 pandemic on March 2020 [19,20]. Considering the wide-ranging adverse consequences of IPV and the persistently high prevalence, there is a need for an integrated and comprehensive approach to the IPV phenomenon in order to align prevention, support and treatment for women at risk of violence. According to Dutton’s nested ecological model [21], the factors associated with the IPV can be considered at four different levels of one’s environment: the macrosystem (cultural beliefs, attitudes and laws), the exosystem(social structures like the work environment or friendships), the microsystem (immediate environment in which the violence occurs), the ontogenetic (individual characteristics of the victim with their beliefs, attitudes and predispositions). A recent meta-analysis examining risk markers for physical IPV victimization found that exosystemic factors (e.g., relationship status, education level) were among the weakest markers of IPV victimization risk, the microsystem factors (such as previous IPV perpetration, previous injury caused by perpetrator, emotional IPV victimization, sexual IPV victimization, emotional IPV perpetration and stalking victimization) were among the strongest risk markers for IPV victimization and factors at the ontogenetic level (depression, PTSD, alcohol use, threats of self-harm and borderline personality disorder) had moderate effect sizes for IPV victimization [22]. At the macrosystem level, cognitive risk factors were identified, such as cultural beliefs about the subordination of women to men, inequality of gender roles and social acceptance of violence against women in an intimate relationship [23]. Although the importance of risk factors at the macrosystem level is crucial to a full understanding of the IPV phenomenon, conclusions from this level lead to social and legal rather than clinical implications [24]. Significant risk factors for IPV victimization at the exosystem and microsystem levels have been extensively described in the literature, but at the ontogenetic level, individual cognitive characteristics have received relatively little theoretical and research attention, despite their potential modifiability [25].

A significant body of research examining factors related to intimate partner violence suggests that experiencing abuse and neglect in childhood increases the risk of IPV victimization in adulthood [26].The cognitive literature has explored the role of cognitive vulnerabilities in the relationship between early negative experiences and subsequent victimization [27]. Previous studies have shown preliminary support for an association between early maladaptive schemas and IPV victimization; however, the evidence base is still small [28,29,30].Schema theory posits that the persistent neglect of core emotional needs early in life creates a particular susceptibility to the development of EMS [31]. Young conceptualized EMS as “a broad, pervasive theme or a pattern, comprised of memories, emotions, cognitions, and bodily sensations, regarding oneself and one’s relationships with others, developed during childhood or adolescence, elaborated throughout one’s lifetime, and is dysfunctional to a significant degree” [32]. Originally, 18 schemas grouped into five main domains were identified. Each schema domain corresponds to a specific unmet need from childhood: disconnection and rejection—lack of stability, safety and emotional care; impaired autonomy and performance—lack of autonomy, sense of competence and sense of identity; impaired limits—lack of realistic limits and self-control; other-directedness—lack of freedom in expressing emotions and needs; and over-vigilance and inhibition—lack of possibility to play, be spontaneous and relax [32]. Research has shown that EMS are associated with a variety of mental health issues, which is consistent with the theoretical background. Empirical evidence has confirmed significant correlations between early maladaptive schemas and personality disorders [33], depression [34,35], eating disorders [36], anxiety disorders, obsessive–compulsive disorder and post-traumatic stress disorder [37]. According to research, EMSs have been found to have a moderate and positive association with interpersonal problems [38]. In light of these findings, cognitive research aimed to investigate the links between EMS and intimate partner violence. A recent meta-analysis of nine studies found that IPV victimization was moderately associated with the disconnection and rejection and impaired autonomy domains and had a small association with other-directedness [25]. However, due to the limited evidence base, additional research is needed to verify the etiological hypothesis of an association between early maladaptive schemas and the subsequent risk of IPV victimization. Schema therapy is considered an effective method of changing the schemas and symptoms of personality disorders [39]. Therefore, a thorough understanding of the role of EMS in IPV victimization might underpin prevention and intervention efforts to break the cycle of violence.

The process of change in the course of therapeutic intervention is dictated by numerous factors. Growing research is exploring the role of various metacognitive abilities in achieving mental well-being, considering metacognitive assessments as promising indicators for identifying current and future mental health conditions [40]. The ability to understand one’s own and others’ mental states is a key factor underlying social interactions and interpersonal relationships [41]. Metacognition is also a component of emotional intelligence, distinguished as strategic emotional intelligence, which includes the abilities related to understanding emotions, cognitive control of emotions and conscious emotions regulation [42]. Early maladaptive schemas activated in adulthood can lead to misinterpretation of social and interpersonal cues, triggering difficult emotions and maladaptive behaviors, and thus contribute to interpersonal problems and perpetuate the original maladaptive schemas [32]. Accordingly, metacognitive abilities, when limited, are believed to be an essential part of both the origination and maintenance of early maladaptive schemas. Indeed, lower levels of emotional intelligence have been found to be associated with a greater likelihood of maladaptive coping in response to EMS [43].

Within this framework, in this study, we put forward a hypothesis that, after accounting for the influence of sociodemographic variables, the domains of EMS would provide additional information for predicting female IPV victimization and non-victimization in a hierarchical manner. Hence, we expected that EMS domains of disconnection and rejection, impaired autonomy and other-directedness would be significant predictors of IPV victimization, which is in line with previous research [25]. In addition, we hypothesized that strategic emotional intelligence would be associated with more severe EMS, in particular, the disconnection and rejection schema domain.

## 2. Materials and Methods

### 2.1. Participants

The total sample consisted of 48 female survivors of intimate partner violence and 48 age-matched women who volunteered to participate in the study and had no prior experience with IPV. A total of 64 female victims of IPV initially agreed to participate in the study group; however, 10 participants withdrew from the study while completing the questionnaires, and another 6 were excluded due to incomplete data. The women gave two main reasons for not participating in the study: strong emotions related to completing the Young Schema Questionnaire-Short Form 3 (YSQ-SF3)and the difficulty level of The Emotional Understanding Test (TRE). All 96 female participants were mothers with full parental rights to at least one of their children. The ages of the women ranged from 20 to 55 years, with the mean age of mothers in the study group being M = 32.91 ± 7.79 and in the control group being 34.27 ± 2.90. Participants from the IPV group were predominantly of lower educational status (77.08%), single, divorced or in an informal relationship, and lived in a large city (64.58%). The reference group was predominantly women with higher education (87.55%), married (81.25%) and living in a large city (56.255). During childhood, 58.33% of women in the IPV group and 2.08% of women in the non-IPV group experienced physical/emotional violence. Additionally, 85.42% and 10.42%, respectively, had witnessed domestic violence in childhood. Women from the IPV group more often reported a psychiatric diagnosis (mainly depression and stress-related disorders). Detailed socio-demographic characteristics of the groups are presented in Table 1.

### 2.2. Measures

The demographics form was developed to suit the specific objectives of the study and included items related to socio-economic factors and participants’ experiences of violence.

Young Schema Questionnaire-Short Form 3 (YSQ-SF3) was used to examine EMS. The scale contains 90 items, each of which is rated on a 6-point Likert-type scale (1 = entirely untrue of me, 6 = describes me perfectly) and higher scores indicate greater severity of EMS(scores range from 5 to 30). YSQ-SF3 evaluates 18 different maladaptive schemas, which are grouped into five domains: disconnection and rejection; impaired autonomy and performance; impaired limits; other-directedness; and over-vigilance and inhibition (as cited in Young et al., 2006 [29]). The Polish adaptation study showed acceptable internal consistency with Cronbach’s alpha ranging from 0.62 (entitlement/grandiosity) to 0.81 (failure) and 0.96 for the overall score [44].

The Emotional Understanding Test (Polish: Test Rozumienia Emocji—TRE) by Matczak and Piekarska was used to evaluate abilities to understand emotions [45]. This test consists of 30 items grouped into 5 subtests (6 items each). The subtests contain a different task related to understanding emotions: (1) ordering emotional states depending on the degree of their intensity; (2) finding the opposite emotion; (3) indicating a simple emotion that makes up a complex emotion; (4) matching the emotional state to the described situation; and (5) indicating conditions that make certain emotional reactions appear in certain situations. The total score in the TRE test is calculated by summing up the points obtained in the 5 subtests (in the range of 0 to 30). The test’s reliability estimated with Cronbach’s α was equal to or higher than 0.78.

### 2.3. Procedure

Participants in the study group were recruited in cooperation with organizations that provide multidimensional help to victims of domestic violence. Specialists announced information about the possibility of participating in the study during group psychoeducational meetings and psychological workshops conducted by support organizations. All of the women in the study group were separated from the perpetrators of violence for at least a month. Women qualified for the research group were under therapeutic care; the inclusion criterion was the stability of their mental state as assessed in consultation with a qualified therapist (licensed psychologist, psychotherapist or crisis intervention specialist). Participants in control group were recruited with use of social media, parenting portals and the snowball method. The average time to complete the test set was 50 min. Data were collected from April 2022 to March 2023. All participants provided signed informed consent to participate in the study. We obtained the ethical approval by the Bioethics Committee at the Medical University of Lodz (RNN/18/KE 12 June 2018) before data collection.

## 3. Results

### 3.1. Differences in Early Maladaptive Schemas between Women with and without IPV History

The characteristics of the measures along with the means, standard deviations and the results of the examination of intergroup differences using the U value together with the level of significance are presented in Table 2 and Table 3. Women who were victims of IPV scored significantly higher on 13 schemas and 4 of the 5 schema domains (Figure 1). The strongest differences concerned the disconnection and rejection domain schemas of emotional deprivation and mistrust/abuse. Other EMS that strongly differentiate women with and without IPV experience are negativity/pessimism, defectiveness/shame, and vulnerability to harm and abandonment.

### 3.2. The Relationship between Early Maladaptive Schemas and Strategic Emotional Intelligence

In order to identify associations between the variables we used Spearman’s correlation coefficients. As shown in Table 4, there are several negative correlations between strategic emotional intelligence scores and individual EMS severity (emotional deprivation; mistrust/abuse; vulnerability to harm; negativity/pessimism; insufficient self-control/self-discipline; and self-sacrifice); nevertheless, the strength of the identified correlations is weak.

Figure 2 illustrates the relationship between the disconnection and rejection schema domain and strategic emotional intelligence. There is a significant and negative relationship between the disconnection and rejection domain and strategic emotional intelligence, which means that an increase in scores on the D/R domain is accompanied by a decrease in scores on strategic emotional intelligence.

### 3.3. Predictors of Intimate Partner Victimization

A three-step hierarchical logistic regression analysis was performed to identify variables related to intimate partner violence. Before the predictors were introduced to the model, the overall prediction rate assumed the value of the probability level (50%). In the first step, socio-demographic variables (age, place of residence, marital status, and employment status) were introduced as control variables into the model, which increased the overall correct classification rate to 81.3, which was statistically significant (χ2(8) = 53.469, *p* < 0.001). Nagelkerke’s R2 was 0.569, demonstrating a satisfactory fit of the model to the current clustering. The introduction of psychiatric diagnosis history information in the second step of the regression analysis showed no improvement in the overall correct classification rate, but improved the fit between prediction and actual clustering, bringing Nagelkerke’s R2 to 0.612. The accuracy of the entire model also improved (χ2(9) = 59.018, *p* < 0.001). When EMS domains were entered in the third step of the analysis, the overall correct classification rate increased to 90.6, which proved statistically significant (χ2(5) = 30.948, *p* < 0.001 for the step, and χ2(14) = 89.966, *p* < 0.001 for the entire model). Nagelkerke’s R2 was 0.811, indicating a high match of the model to the current clustering. As shown in Table 5, the last model revealed that demographic variables, i.e., marital and employment status, and two EMS schema domains (disconnection and rejection and impaired limits) were significant predictors of IPV victimization. Specifically, being in an informal relationship and being unemployed were risk factors for experiencing IPV victimization. Regarding the test variables, for each one-unit increase in the score for the disconnection and rejection schema domain, the odds of belonging to the IPV group increase by 16% (B = 0.145, *p* = 0.010). On the other hand, for each one-unit increase in the score for the impaired limits schema domain, the odds of belonging to the IPV victim group decreased by 18% (B = −0.194, *p* = 0.028).

## 4. Discussion

To summarize, our study aimed to investigate cognitive–affective vulnerabilities to IPV victimization. Broadly, the results of our study revealed that the schema domains disconnection and rejection and impaired limits were significant predictors of IPV victimization, but did not support the predictive value for impaired autonomyandother-directedness. The increase in the severity of disconnection and rejection scores was accompanied by a decrease in strategic emotional intelligence.

Given the associated socio-demographic factors of IPV, marital status and employment status were powerful predictors of further victimization, which is generally in line with previously identified risk factors for IPV. Recent meta-analysis assessed risk markers for IVP, indicating exosystem factors (e.g., financial stress, lower levels of education, not being married, and higher number of children), microsystem factors (e.g., child abuse in family of origin, witnessing IPV in family of origin) and ontogenetic factors (e.g., PTSD, depression) that were positively associated with IPV victimization [22]. These factors strongly differentiate the sociodemography of our research and reference group; however, due to the sample size, not all of them could be controlled in the logistic regression analysis due to the risk of potential issues in the model.

The results of our study revealed significant differences in the severity of EMS between women with and without a history of IPV. These differences were more complex and massive than previously reported in the studies, which may be explained by differences in sociodemographic variables between the study group and the control group. There are two other studies on the relationship between EMS and IPV victimization that used a case–control design [29,30]. The research sample in the Pietri and Bonnet study was recruited from a housing unit for distressed mothers and children, then divided into groups of women with and without a history of IPV. The study population was also expanded through recruitment through the university and professional network; however, there is no information on the detailed characteristics of the compared groups [29]. The sampling in the Pietri and Bonnet study is somewhat similar to our recruitment strategy. We recruited women for the study group, among others, in a Single Mother’s Home. After in-depth consultations with the institution’s specialists, we decided not to include women from Single Mother’s Home in the control group, even if the declared reason for seeking shelter was not violence. The experience of social therapists shows that the overwhelming number of women sheltering in a Single Mother’s Home experienced, to some extent, violent behavior by their intimate partners. Due to the normalization of aggressive behavior in the basic social environment of these women, some gained insight into the experience of violence only during therapy. Considering the above, similarly to Taşkale and Soygüt [30], we recruited participants to the reference group using the snowball method; finally, the reference group and the study group were similar in terms of age, but not in terms of education. Regardless of the differences in sample selection, in line with the results of the aforementioned researchers, the disconnection and rejection domain predicted the risk of exposure to IPV victimization, while being the most severe domain in the study group. These findings align consistently with the underlying assumptions of Young’s schema theory [32]. The disconnection and rejection domain is considered the most harmful and destructive for the individual, having its sources in adverse and traumatic childhood experiences [32]. This theoretical framework is reflected in the research. A recent meta-analysis by Pilkington et al. [25] found that the disconnection and rejection domain have the strongest correlations with incidents of IPV victimization. Moreover, the D/R domain has also been shown to mediate the link between childhood abuse and neglect and subsequent victimization of IPV [28,46,47]. As such, women who report having experienced IPV tend to expect that their fundamental emotional needs for safety, protection, stability, empathy, acceptance and respect will not be met by others, particularly their romantic partners. Within this domain, there were two schemas that most differentiated the group of women affected by IPV from the control group: emotional deprivation and mistrust/abuse. These schemas were also significantly higher among women who had experienced and/or witnessed violence as children. Similar results were obtained in a study by other researchers where these schemas contributed to IPV victimization [28,29]; however, in a recent meta-analysis, only the mistrust and abuse scheme received enough data to be considered a predictor of IPV victimization [25]. Additionally, regression analysis results showed an inverse relationship between the impaired limits schema domain and the likelihood of experiencing IPV victimization. As previously investigated, the impaired limits domain correlates with aggressiveness [48] and violence perpetration [49]. Considering the severity of the impaired limits domain dimensionally, it can be assumed that extremely low scores may be associated with submissiveness, suppression of one’s own needs, low self-esteem and difficulty in recognizing one’s rights, extremely high scores may be associated with impulsiveness, disregard for the rights of others, inability to cooperate, entitlement, and with moderate scores, it can be associated with assertiveness and good self-opinion.

Our results showed that lower levels of strategic emotional intelligence were associated with more severe activation of 6 out of 18 EMS and that stronger activation of the D/R domain was accompanied by lower levels of strategic emotional intelligence. To the best of our knowledge, there is only one study to date on the relationship between emotional intelligence and EMS, the results of which are generally consistent with our research findings [43]. The emotionality domain of trait emotional intelligence (TEI), measuring the ability to perceive one’s own and other people’s emotions and the ability to express and communicate emotions, thus sharing the objectives of the TRE measurement, showed weak or moderate negative correlations with all schema domains except other-directedness. In our study, the strongest correlations with strategic emotional intelligence showed two maladaptive schemas (emotional deprivation and mistrust/abuse) from the disconnection and rejection domain (r = −0.29, *p* < 0.05). Our findings suggest that when individuals do not receive adequate support, love, acceptance and validation of other basic emotional needs and, consequently, develop beliefs that in close relationships basic emotional needs cannot be met, and expectations that people are prone to harm, abuse and humiliate may experience difficulties in understanding, managing and expressing their emotions. The domain of disconnection and rejection is also closely related to exposure to adverse events in childhood [50]. As previously reported, adverse childhood events have a detrimental effect on emotional intelligence [51]. Therefore, people who in childhood did not experience the warmth, protection and involvement of parents may consequently become more susceptible to developing early maladaptive schemes from the domain of disconnection and rejection, and in adulthood may experience greater difficulties in the socio-emotional sphere and encounter with interpersonal problems [38]. Thus, taking into account that the disconnection and rejection domain may be associated with the experience of violence and may be accompanied by difficulties in understanding, adequately perceiving and expressing emotions, may help in targeting therapeutic interventions (e.g., implementing psychoeducation, using imagery techniques and chair work) in alleviating suffering from traumatic experiences. The current study has some limitations that should be acknowledged. First, the sample size in this study was relatively small, which may have resulted in limited statistical power in regression analyses. Therefore, future research should include a larger population to increase statistical power and facilitate more robust conclusions. There are inherent challenges and complexities in conducting research with a survivors of violence Although participants were assured of full anonymity and no impact on the care provided in support institutions, of the more than 100 women experiencing IPV invited to participate in the study, only 48 returned completed questionnaires. In our study, one of the most severe schemas among victims of violence was mistrust/abuse. Therefore, women who have experienced violence can expect intentional harm, punishment, humiliation or abuse, becoming more suspicious and willing to withdraw or hide their experiences, emotions and beliefs. This, in turn, may affect the sincerity of the responses given. Further research involving a mixed quantitative and qualitative methodology is strongly encouraged. What is more, the current design of the study does not allow conclusions to be drawn as to the direction of the relationship between the variables. Further longitudinal studies are needed to assess the direction of the observed relationships. Finally, it should be mentioned that the compared groups were not homogenous in terms of some demographic variables, which may have distorted the nature of the observed differences. A valuable solution for further research may be a close match of the reference group in terms of socio-demographic factors.

In summary, notwithstanding the aforementioned limitations, the results of our research broaden the view on cognitive–affective risk factors of IPV and highlight the crucial role of EMS on the psychological well-being of women affected by IPV. Our findings add to the emerging evidence of a link between disconnection and rejection domain and IPV victimization. As a consequence, maladaptive beliefs that interpersonal relationships are unstable and insecure and expose to the risk of humiliation and harm, and that basic emotional needs cannot be satisfied in close relationships are associated with a higher risk of intimate partner violence. Moreover, lower levels of strategic emotional intelligence were associated with more severe EMS, especially schemas of emotional deprivation, mistrust/abuse and vulnerability to harm. Longitudinal research is needed to determine the role of strategic emotional intelligence in perpetuating early maladaptive schemas. Since violence against women is a serious threat to well-being at the individual, family and societal levels, encouraging a multidisciplinary approach in prevention and support is crucial to mitigate the devastating effects of IPV victimization. As schema therapy is a recognized therapeutic method that effectively alleviates the severity of schemas [52], it is worth considering it as an important method of supporting victims of IPV violence.

## Figures and Tables

**Figure 1 brainsci-13-01118-f001:**
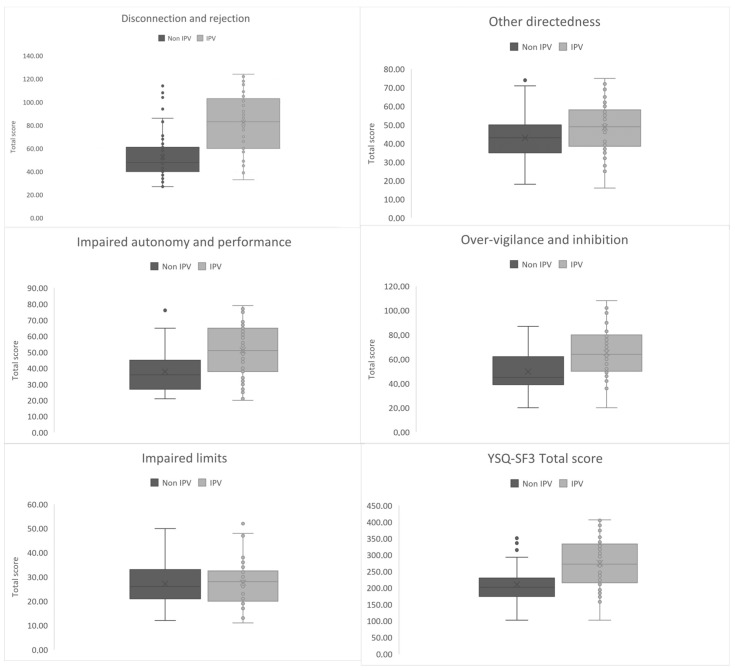
A box and whisker plot of EMS’s domain scores of study vs. reference group. The gray center lines represent the median value, while the gray boxes contain the 25th to 75th percentiles of the dataset. Gray whiskers indicate the 5th and 95th percentiles, values beyond the upper and lower bounds are considered outliers, marked with grey dots.

**Figure 2 brainsci-13-01118-f002:**
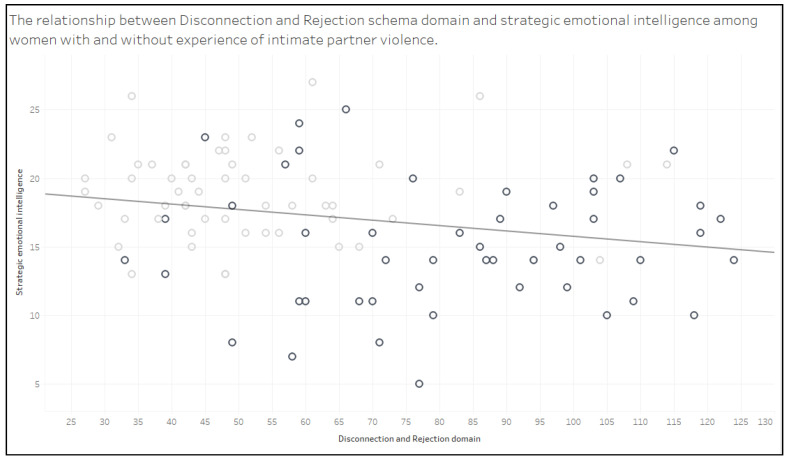
Scatterplot illustrating the relationship between disconnection and rejection schema domain and strategic emotional intelligence. The light gray spots represent results of women in the reference group, while the dark gray spots represent results of women in the study group. Equation: strategic emotional intelligence = −0.0386897 × disconnection and rejection domain + 19.671; trend line: *p*-value: 0.017.

**Table 1 brainsci-13-01118-t001:** Socio-demographic characteristics of participants.

Variables	IPV * Group		Non-IPV * Group	
	*n*	%	*n*	%
Age M ± SD	32.91 ± 7.79		34.27 ± 2.90	
Range	20–55		28–42	
Education				
Primary	18	37.5	-	-
Vocational	6	12.5	-	-
Secondary	13	27.08	6	12.5
Higher	11	22.92	42	87.5
Marital status				
Marriage	17	35.42	39	81.25
Informal relationship	9	18.75	7	14.58
Divorced	5	10.42	2	4.17
No relationship	17	35.42	-	-
Place of residence				
Countryside	2	4.17	14	29.17
City up to 50,000 residents	4	8.33	4	8.33
City 50,000–100,000 residents	11	22.92	3	6.25
City over 100,000 residents	31	64.58	27	56.25
Children				
1	18	37.5	27	56.25
2	16	33.33	20	41.67
3	14	29.17	1	2.08
Childhood victimization	28	58.33	1	2.08
Witnessing violence in childhood	41	85.42	5	10.42
Psychiatric diagnosis	23	47.92	9	18.75

* IPV = intimate partner violence.

**Table 2 brainsci-13-01118-t002:** Comparison of the measures.

Measures	IPV Group			Non-IPV Group			U	Z	*p*
	Mean	St.Dev	Rank Sum	Mean	St.Dev	Rank Sum			
YSQ-SF3 *									
Emotional deprivation	17.69	6.42	3200.5	8.83	4.37	1455.5	279.5	−6.389	0.000
Abandonment	18.25	6.55	2913.0	12.67	4.76	1743.0	567.0	−4.283	0.000
Mistrust/Abuse	19.06	5.70	3118.5	10.98	5.24	1537.5	361.5	−5.789	0.000
Social isolation/ Alienation	15.50	6.20	2857.5	10.67	5.60	1798.5	622.5	−3.877	0.000
Defectiveness/ Shame	12.98	6.63	2984.5	7.50	4.00	1671.5	495.5	−4.807	0.000
Failure to achieve	13.73	6.18	2800.5	9.67	5.52	1855.5	679.5	−3.459	0.000
Dependence/ Incompetence	11.69	5.46	2771.5	8.15	3.22	1884.5	708.5	−3.246	0.000
Vulnerability to harm	16.48	6.36	2975.5	10.23	4.21	1680.5	504.5	−4.741	0.000
Enmeshment/ Undeveloped self	9.77	4.63	2516.0	8.15	2.92	2140.0	964.0	−1.374	0.169
Entitlement/Grandiosity	13.00	4.01	2369.0	12.71	4.34	2287.0	1111.0	−0.297	0.766
Insufficient self-control/ Self-discipline	14.21	5.76	2422.0	13.25	5.09	2234.0	1058.0	−0.685	0.492
Subjugation	13.29	5.72	2781.0	9.41	4.56	1875.0	699.0	−3.316	0.000
Self-sacrifice	21.04	6.23	2740.5	17.50	5.32	1915.5	739.5	−3.019	0.003
Approval-seeking/ Recognition-seeking	13.83	5.26	2265.5	14.29	5.26	2390.5	1089.5	−0.454	0.649
Emotional inhibition	14.50	6.43	2781.5	10.29	5.35	1874.5	698.5	−3.319	0.001
Unrelenting standards/ Hypercriticism	17.31	5.40	2581.0	15.40	4.95	2075.0	899.0	−1.850	0.064
Negativity/Pessimism	19.06	6.71	2994.0	12.02	5.23	1662.0	486.0	−4.877	0.000
Punitiveness	14.85	5.34	2884.0	10.50	3.92	1772.0	596.0	−4.071	0.000

* YSQ-SF3 = Young Schema Questionnaire-Short Form 3.

**Table 3 brainsci-13-01118-t003:** Intergroup comparison of EMS’s domains.

YSQ-SF3 *	IPV Group	Non-IPV Group	U	Z	*p*
Schema Domain	Rank Sum	Rank Sum			
Disconnection and rejection	3572.5	1887.5	456.5	−5.82	0.000
Impaired autonomy and performance	3340.5	2119.5	688.5	−4.31	0.000
Impaired limits	2765.5	2694.5	1263.5	−0.57	0.569
Other directedness	3088.5	2371.5	940.5	−2.67	0.007
Over-vigilance and inhibition	3387.0	2073.0	642.0	−4.61	0.000

* YSQ-SF3 = Young Schema Questionnaire-Short Form 3.

**Table 4 brainsci-13-01118-t004:** Correlation matrix for tested variables.

Variable	1	2	3	4	5	6	7	8	9	10	11	12	13	14	15	16	17	18	19
1.TRE ^1^	1.00																		
2. ED ^2^	**−0.29**	1.00																	
3. Ab ^3^	−0.14	**0.58**	1.00																
4. M/A ^4^	**−0.28**	**0.66**	**0.69**	1.00															
5. Si/A ^5^	−0.05	**0.59**	**0.50**	**0.62**	1.00														
6. D/S ^6^	−0.19	**0.69**	**0.65**	**0.59**	**0.70**	1.00													
7. FA ^7^	−0.09	**0.55**	**0.53**	**0.56**	**0.65**	**0.69**	1.00												
8. D/I ^8^	−0.19	**0.54**	**0.55**	**0.54**	**0.49**	**0.59**	**0.68**	1.00											
9. Vh ^9^	**−0.29**	**0.69**	**0.63**	**0.70**	**0.55**	**0.66**	**0.55**	**0.61**	1.00										
10. E/Us ^10^	−0.08	**0.28**	0.15	0.13	0.08	**0.33**	**0.24**	**0.32**	**0.28**	1.00									
11. Sb ^11^	−0.07	**0.60**	**0.61**	**0.54**	**0.60**	**0.69**	**0.58**	**0.66**	**0.67**	**0.33**	**1.00**								
12. Ss ^12^	**−0.21**	**0.47**	**0.57**	**0.58**	**0.43**	**0.39**	**0.44**	**0.41**	**0.56**	0.08	**0.49**	1.00							
13. Ei ^13^	−0.05	**0.56**	**0.53**	**0.55**	**0.71**	**0.67**	**0.57**	**0.49**	**0.53**	0.12	**0.63**	**0.40**	1.00						
14. Us/H ^14^	0.00	**0.46**	**0.38**	**0.37**	**0.43**	**0.38**	**0.50**	**0.37**	**0.41**	0.04	**0.48**	**0.47**	**0.42**	1.00					
15. E/G ^15^	−0.17	0.13	0.15	0.19	0.15	0.07	0.14	0.17	0.16	0.09	0.01	**0.26**	0.07	**0.38**	1.00				
16. Is/Sd ^16^	**−0.21**	**0.29**	**0.35**	**0.24**	**0.28**	**0.41**	**0.41**	**0.53**	**0.46**	0.20	**0.47**	**0.37**	**0.34**	**0.38**	**0.32**	1.00			
17. As/Rs ^17^	−0.11	0.14	**0.33**	0.15	0.09	0.18	**0.36**	**0.30**	**0.22**	**0.21**	**0.21**	**0.21**	0.15	**0.42**	**0.51**	**0.51**	1.00		
18. N/P ^18^	**−0.25**	**0.73**	**0.66**	**0.74**	**0.59**	**0.68**	**0.64**	**0.60**	**0.87**	**0.24**	**0.64**	**0.62**	**0.56**	**0.58**	0.13	**0.50**	**0.29**	1.00	
19. P ^19^	−0.16	**0.55**	**0.57**	**0.57**	**0.48**	**0.53**	**0.59**	**0.46**	**0.50**	0.15	**0.47**	**0.53**	**0.40**	**0.55**	**0.23**	**0.36**	**0.35**	**0.63**	1.00

^1^ TRE = The Emotional Understanding Test (Polish Test Rozumienia Emocji); ^2^ emotional deprivation; ^3^ abandonment; ^4^ mistrust/abuse; ^5^ social isolation/alienation; ^6^ defectiveness/shame; ^7^ failure to achieve; ^8^ dependence/incompetence; ^9^ vulnerability to harm; ^10^ enmeshment/undeveloped self; ^11^ subjugation; ^12^ self-sacrifice; ^13^ emotional inhibition; ^14^ unrelenting standards/hypercriticalness; ^15^ entitlement/grandiosity; ^16^ insufficient self-control/self-discipline; ^17^ approval-seeking/recognition-seeking; ^18^ negativity/pessimism; ^19^ punitiveness; marked correlation for *p* < 0.05.

**Table 5 brainsci-13-01118-t005:** Last step of the hierarchical logistic regression model predicting IPV victimization.

	B	Wald	df	*p*	Exp(B)	95% C.I. for EXP(B)	
						Lower	Upper
Age	0.155	2.390	1	0.122	1.167	0.959	1.421
Place of residence		2.780	3	0.427			
Place of residence (1)	−0.270	0.013	1	0.909	0.764	0.008	76.258
Place of residence (2)	2.663	2.405	1	0.121	14.336	0.495	414.969
Place of residence (3)	1.442	1.143	1	0.285	4.229	0.301	59.453
Marital status		5.930	2	0.052			
Marital status (1)	2.938	4.793	1	0.029	18.874	1.360	261.879
Marital status (2)	2.638	3.284	1	0.070	13.987	0.806	242.625
Employment status		8.129	2	0.017			
Employment status (1)	4.874	7.287	1	0.007	130.907	3.801	4508.158
Employment status (2)	2.170	3.662	1	0.056	8.760	0.949	80.866
History of psychiatric diagnosis (1)	0.852	0.721	1	0.396	2.345	0.328	16.778
Disconnection and rejection *	0.145	6.655	1	0.010	1.156	1.036	1.291
Other-directedness *	−0.029	0.264	1	0.607	0.971	0.870	1.085
Impaired autonomy and performance *	−0.001	0.000	1	0.992	0.999	0.888	1.125
Over-vigilance and inhibition *	−0.041	0.423	1	0.516	0.960	0.850	1.085
Impaired limits *	−0.194	4.830	1	0.028	0.823	0.692	0.979
Constant	−9.085	3.614	1	0.057	0.000		

* YSQ-SF3 = Young Schema Questionnaire-Short Form 3.

## Data Availability

The data presented in this study are available on request from the corresponding author. The data are not publicly available due to privacy issues.

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
