# Peer review of "Cognitive–Affective Risk Factors of Female Intimate Partner Violence Victimization: The Role of Early Maladaptive Schemas and Strategic Emotional Intelligence"

_brainsci, 2023, doi:10.3390/brainsci13071118_

Round 1

Reviewer 1 Report

The study is very interesting and insightful.

Please take note of the following comments:

Lines 66-67: "the existing IPV theoretical framework still has a relatively low effectiveness in reducing IPV perpetration"

Please provide a description of the existing IPV theoretical framework.

Lines 72-74: "A significant body of research examining factors related to intimate partner violence suggests that experiencing abuse and neglect in childhood increases the risk of IPV exposure in adulthood."

Please specify whether the risk of exposure is for victims or perpetrators? 

Lines 76-77: "Previous research on cognitive factors in IPV has highlighted the role of early 76 maladaptive schemas (EMS) in relation to victimization." 

In light of the above statement, the gap in the existing research as a rationale for this study is not clear.

Lines 93-94: "According to research, EMSs have been found to have a moderate and positive association with interpersonal problems."

Support for the above statement is needed.

Lines 134-135: "10 participants withdrew from the study while completing the questionnaires, another 6 were excluded due to incomplete data."

Please specify whether the participants withdrew from the study group or control group? In addition, clarify the number of participants who did participate.  

There are just a few minor errors. For example, Lines 181-182: "Participants in control group were recruited with use of social media, parenting portals, the snowball method." and between portals and the snowball method is missing. 

Reviewer 2 Report

Thank you very much for giving me the opportunity to review this manuscript. The idea of your article is interesting, my recommendations are the following:

Abstract

it would be recommended to review the conclusions because it is not connected with the title of the work.

There are expressions that creak, like "Still little is known" that doesn't sound like English but rather a machine translation

 Introduction

There is a lack of talking specifically about vulnerability. In victimology, the presence of early risk factors that generate greater chances of suffering victimization is vulnerability.

As for the schemes, it is not specified what type. What the literature exposes are sexist cognitive distortions, of gender roles regarding the role of women, of enduring violence due to low self-esteem... It is necessary to specify because it is a field of broad variables and in general it would not be adequate.

A methodological flaw in the hypothesis is that emotional intelligence is closely related to its development by the environment. When the environment is private, it does not develop as it could be a liquid and non-static capacity. Likewise, from what protection or what influence? From seeing predators and not having relationships? Knowing how to detect relationships of abuse? This point is still not specified.

Potentially, for intelligence to discriminate between victims and non-victims seems audacious, leaving behind the part of the aggressor, and making the victims responsible in capacities. And literature does not follow this line, nor is it based on why it should go this way now.

Methods

In procedure the training system for women is not developed (announcement, calls...)

Results

It is assumed that violence means having less emotional intelligence, when the consequences of violence are lower levels. The causality is not clear here, nor does the literature support it. If we measure levels of emotional intelligence in childhood we will not predict violence or not. Rather, measuring violence implies lower levels of emotional intelligence development.

Nor are cases of resilience established, where regardless of the victimization experienced, long-term sequelae are not generated.

Risk factors are not discriminated from those that entail sequelae. When victimization is suffered, there is psychological damage that produces vulnerability. But these consequences imply vulnerability that generates a greater probability of suffering violence. But the risk factor is the previous victimization, and the sequels are the consequences. There is no clarity in discriminating from the psycho-emotional profile of the victim, and it is necessary to clarify an IPV map with precision.

Conclusion

It would be advisable to develop conclusions
